# FIELDWISE FACTORIZED NETWORKS FOR TABULAR DATA CLASSIFICATION

## ABSTRACT

Tabular data is one of the most common data-types in machine learning, however, deep neural networks have not yet convincingly outperformed classical baselines on such datasets. In this paper, we first investigate the theoretical connection between neural network and factorization machine techniques, and present *fieldwise factorized neural networks* (F2NN), a neural network architecture framework that is aimed for tabular classification. Our framework learns high-dimensional field representations by a low-rank factorization, and handles both categorical and numerical fields. Furthermore, we show that simply by changing our penultimate activation function, the framework recovers a range of popular tabular classification methods. We evaluate our method against state-of-the-art tabular baselines, including tree-based and deep neural network methods, on a range of tasks. Our findings suggest that our theoretically grounded but simple and shallow neural network architecture achieves as strong or better results than more complex methods.

## 1 INTRODUCTION

Deep learning methods have recently made extraordinary progress by using architectures that utilize the regular structure of visual or linguistic data. However, one of the most common type of data consists of tables with arbitrarily ordered fields (columns). Such *tabular* data are unable to exploit neural architectures with strong structural inductive biases. In fact, deep learning methods have not been able to convincingly outperform classical baselines on such datasets. As such, traditional machine learning methods, such as Gradient-Boosted Decision Trees (Friedman (2000)) and Factorization Machines (Rendle (2010); Zhang et al. (2018); Pan et al. (2018); Sun et al. (2021)), still dominate even against recently proposed, specialized neural architectures (Arik & Pfister, 2021; Yoon et al., 2020).

In this paper, we first investigate the connection between shallow neural networks and factorization machine techniques, which are a dominant approach for solving benchmark tabular classification tasks, such as click-through-rate (CTR) prediction and user-movie recommendations. Based on this theoretical connection, we propose a neural network-based framework: *fieldwise factorized neural networks* (F2NN). Our framework is general, and is aimed for tabular classification. By changing a single activation layer in F2NN, our framework transforms between being a factorization machine, a shallow ReLU network or a wide-and-deep model. Our framework consists of a set of fieldwise wide yet shallow neural networks, which are aggregated by summation and passed through non-linearity and classification layers. Differently from typical factorization machine methods, our framework can handle numeric and categorical data as well as multiclass outputs.

Our framework is characterized by two main components: i) learning a high-dimensional representation per-field by factorized networks, ii) an adaptive activation layer acting on the representation aggregated from all fieldwise networks. Although the theoretical analysis shows that high-dimensional fieldwise embeddings are desired, mapping the high-dimensional features to them requires huge linear layers. Factorizing these layers using low-dimensional layers makes training such fieldwise networks feasible. We also show that for these fieldwise factorized networks, width is preferable to depth. Another key insight is that frameworks with different post-aggregation activation functions can be suitable for different types of tasks. While quadratic activations are optimal for user-product recommendations, ReLU networks are sometimes better for fully-numeric tasks. We thus seek a post-aggregation activation that can enjoy the best of both worlds, and demonstrate that GELU

(Hendrycks & Gimpel, 2016) satisfies these requirements. Our theoretical analysis shows that the GELU activation behaves as either quadratic and ReLU activations at different input scales and is thus suitable across the full range of tabular tasks. Consequently, GELU can be used as an efficient alternative for exhaustive search over the optimal activation or an ensemble approach.

We present an extensive experimental analysis, evaluating our framework on a wide range of popular numeric and categorical tabular classification tasks as well as CTR prediction. We find that our simple method performs comparably or better than tree-based methods and recent complex deep learning method on general classification, and outperforms strong baselines on CTR prediction. Our results show that the network width and activation play a bigger role than depth or attention for tabular data.

Our contributions include: (1) Establishing a connection between factorization machines and shallow neural networks. (2) Presentation of a general, theoretically grounded framework for tabular data classification. Our method can express many popular methods by simply modifying its activation function. It also makes learning high-dimensional fieldwise representations feasible by low-rank factorization. (3) Extensive evaluation on a range of tabular classification tasks and demonstrating above or comparable to state-of-the-art performance.

## 1.1 RELATED WORK

**Tree-based models:** Ensembles of decision trees, such as GBDT (Gradient Boosting Decision Tree) (Friedman, 2001), are typically the top-choice for tabular data classification . At the moment, there are several established GBDT libraries, such as XGBoost (Chen & Guestrin (2016)), LightGBM (Ke et al. (2017)), CatBoost (Prokhorenkova et al. (2018)), which are widely used by both machine learning researchers and practitioners. Tree-based methods have some limitations that are addressed by other methods such as neural networks: requirement for feature engineering, poor scaling to high input cardinality, reduced representation transferability and GPU acceleration.

**FM-based models:** Click-through rate (CTR) prediction plays a key role in recommendation systems and online advertising. Factorization machine models are a popular family of methods for CTR prediction, including: matrix factorization (Koren (2008)), factorization machine (FM, Rendle (2010)), Field-aware Factorization Machines (FFM, Zhang et al. (2018)), Field-weighted Factorization Machine (FwFM, Pan et al. (2018)) and Field-matrixed Factorization Machines (FmFM, Sun et al. (2021)). We will give a more in-depth introduction to FM-based methods in Sec. 2. As a step toward combining deep neural network for this task, Wide & Deep (Cheng et al. (2016)) proposed to train a joint network that combines a linear model and a deep neural network. DeepFM (Guo et al. (2017)) suggest to learn a low-order feature interactions through the FM component instead of the linear model. Since then, various embedding-based neural networks have been proposed to improve the performance (Deng et al. (2021); He & Chua (2017); Wang et al. (2017); Lian et al. (2018)). Unlike He & Chua (2017), our framework is able to learn higher dimensional representations, as well as leveraging non-linear embeddings for the numerical fields. It is also more general, as it able to recover range of tabular models by changing the activation.

**Fieldwise models:** Li et al. (2020) present a model for categorical data, that utilizes linear models with variance and low-rank constraints, and is also interpretable in a field-wise manner. Although we share the ideas of fieldswise low-rank factorization, our framework allows non-linearity, as well as handles both numerical and categorical fields. Luo et al. (2020) propose NON to take advantage of intra-field information and non-linear interactions. However, our components are much simpler yet effective. Our fieldwise networks rely on low-rank factorization of wider networks and our non-linearity is based on an adaptive activation, while NON utilize only ReLU activation and ensembles multiple heavy aggregation mechanisms, such as attention and deep neural networks.

**Deep models for general tabular data:** While classical methods are still the industry favorite, some recent work propose to use deep learning for tabular data. For example, TabNet (Arik & Pfister (2021)) uses neural networks to mimic decision trees by placing importance on only a few features at each layer, using modified attention layers. Yoon et al. (2020) propose VIME, which employs MLPs in a technique for pre-training based on denoising. Transformer models for more general tabular data include TabTransformer (Huang et al. (2020)), which uses a transformer encoder to learn contextual embeddings only on categorical features. The main issue with this model is that numerical data do not go through the self-attention block, but are only fed to an MLP. SAINT (Somepalli et al. (2021))

address that issue by projecting numerical features to the higher dimensional embedding space and passing them, together with the categorical embeddings, through the transformer blocks. In addition, SAINT propose using attention in the rows level, to explicitly allow data points to attend each other.

**Critiques of deep neural networks for tabular data:** Recently, several works criticized the proposed deep models for tabular data. For example, several works criticize MLP since it is unsuitable for modeling multiplicative interactions between features (Rendle et al. (2020)). In addition, Shwartz-Ziv & Armon (2020) criticize deep tabular data models on claiming to outperform XG-Boost, while their study shows that XGBoost outperforms these deep models across the datasets.

**Deep learning on unordered sets:** Qi et al. (2017) suggest a unified architecture for handling point cloud data. From a data structure point of view, both point cloud and tabular data are an unordered set of vectors. The basic architecture of their network is surprisingly simple as in the initial stages each point is processed identically and independently. In addition, key to this approach is the use of max pooling for aggregating informative points of the point cloud. Our framework architecture shares similar architecture ideas, by the independent processing of each field, followed by aggregation. However, we use a different network for every field while they share the same network to all points (due to the symmetric of point cloud data).

## 2 THE THEORETICAL CONNECTION BETWEEN FACTORIZATION MACHINES AND FIELDWISE NEURAL NETWORKS

In this section, we will briefly overview factorization machine approaches. We will then demonstrate a theoretical connection between them and shallow neural networks. This will form the basis for our final, generalized framework in Sec. 3.

### 2.1 BACKGROUND

In this section we describe a common approach for classification of tabular data. For ease of explanation, we detail the case of *binary* classification of multi-field *categorical* data. However, there is no loss of generality, our final approach applies for categorical and numerical as well as for multi-class tabular datasets.

**Preliminaries:** A training dataset consists of $S$ labelled samples $\{(x^{(1)}, y^{(1)}), (x^{(2)}, y^{(2)})..(x^{(S)}, y^{(S)})\}$ , where $x$ and $y$ are a sample and label respectively. Each sample $x$ is specified by $C$ categorical features $f_1, ..f_C$ and $J$ different fields $F_1, ...F_J$. Each field may contain multiple features, while each feature belongs to only one field. An example of fields are "Country" or "City", whereas features could be "Japan" or "Rome". To simplify the notation, we use index $i$ to represent the feature $f_i$ and $F(i)$ to represent the field which $f_i$ belongs to. $S_F$ denotes the set of features belonging to field $f$ and $J_{F(i)}$ represents their number. We denote $x_i^{(s)} = 1$ if feature $i$ is active for this instance, otherwise $x_i^{(s)} = 0$. We denote the one-hot vector of features per-field $\boldsymbol{x_F} = concat\{x_i \in S_F\}$. We denote the embedding of feature $i$ as $\boldsymbol{e_i} \in \mathbb{R}^K$, where $K$ is the (usually small) feature embedding dimension. We denote by $E_F \in \mathbb{R}^{K \times J_{F(i)}}$ the field embedding matrix whose rows are the embeddings $\boldsymbol{e_i}$ of the features $S_F$ belonging to the field $f$.

**Logistic Regression (LR)** is probably the most widely used model for this task. However, linear models lack the capability to represent feature interactions as cross features are often significant (Chapelle et al. (2014)). *Degree-2 Polynomials (Poly2)* (Chang et al. (2010)) model a general way to address this problem. Such models learn a dedicated weight, $W_{ij}$, for each feature conjunction resulting in a field interaction matrix $W \in \mathbb{R}^{C \times C}$.

**Factorization Machine Models:** Estimating $W$ is hard due to its huge dimensionality and missing values. Factorization-machine models propose to learn the effect of feature conjunctions by low-rank factorization of the interaction matrix $W$. Factorization machine methods approximate the feature interaction strength as the scalar product of their embeddings weighted by a matrix $M_{F(i),F(j)} \in \mathbb{R}^{K \times K}$, that depends on the fields of the two features: $W_{i,j} \approx \boldsymbol{e_i}^T M_{F(i),F(j)} \boldsymbol{e_j}$. Different factorization machine methods are distinguished by their particular choice of matrix $M_{F(i),F(j)}$. We present the choices of $M_{F(i),F(j)}$ taken by four representative factorization ma-

chine methods in Eq. 1: FM (Koren, 2008), FFM (Zhang et al., 2018), FwFM (Pan et al., 2018), FmFM (Sun et al., 2021):

$$M_{F(i),F(j)} = \begin{cases} I_K & \Phi_{FM} \\ P_{F(j)}^T P_{F(i)} & \Phi_{FFM} \\ I_K \cdot r_{F(i),F(j)} & \Phi_{FwFM} \\ Entire\ Matrix & \Phi_{FmFM} \end{cases} \tag{1}$$

Where $I_K$ is the identity matrix, $P_{F(i)}$ is a per-field sparse binary projection matrix, $r_{F(i),F(j)}$ is a scalar value learned for every pair of fields $(F(i), F(j))$. For FM and FFM, $M$ is not learned, while for FwFM a scalar per pair of fields is learned, and for FmFM the entire matrix is learned. See Appendix B for detailed formulation of $M$ for each FM-based model.

## 2.2 FACTORIZATION-BASED MODELS AS A FIELDWISE FACTORIZED NEURAL NETWORK

In this section, we show that factorization machines are a special case of shallow neural networks.

Let us denote the second order interaction term as $q_2 = \sum_i \sum_j x_i x_j \cdot e_i^T M_{F(i),F(j)} e_j$. We demonstrated above that the crux of factorization machine methods is in the representation of the interaction matrix $M_{F(i),F(j)}$. Let $M$ be the matrix that consists of the matrices $M_{F(i),F(j)}$ for all pairs of fields. Since there is no order between the fields, for each pair only one matrix is learned, meaning $M_{F_iF_j} == M_{F_jF_i}$, while the diagonal block matrices are zeros. See Appendix B for a detailed formulation of M. We observe that $M$ is symmetric matrix, and therefore an eigen-decomposition can be applied to it: $M = U^T \Lambda U$. Here we propose to utilize a reduced-rank factorization of $M$ resulting in a field-wise factorization: $M_{F(i),F(j)} = U_{F(i)}^T \Lambda U_{F(j)}$, where $U_{F(i)} \in R^{d \times K}$. Note that the full rank dimension of $M$ is equal to the number of fields multiplied by the number of dimensions per-embedding. Using this factorization the feature interaction term $q_2$ can now be written:

$$q_2 = \sum_i \sum_j (x_i U_{F(i)} e_i)^T \Lambda (x_j U_{F(j)} e_j) \tag{2}$$

We rewrite the sums, as the sum over the field $f$ and the sum over the per-field indices $i \in S_F$:

$$q_2 = \sum_F \sum_{F'} (\sum_{i \in S_F} x_i e_i)^T U_F^T \Lambda U_{F'} (\sum_{j \in S_{F'}} x_j e_j) \tag{3}$$

Note that as $x_F$ is one-hot (only a single element has a non-zero value), the product $E_F x_F = \sum_{j \in S_{F'}} x_j e_j$ can be efficiently computed using an embedding layer $E_F x_F$:

$$q_2 = (\sum_F U_F E_F x_F)^T \Lambda (\sum_{F'} U_{F'} E_{F'} x_{F'}) \tag{4}$$

As the right-hand left-hand vectors are equal, it yields the simple expression:

$$q_2 = diag(\Lambda) \cdot (\sum_F U_F E_F \ x_F)^2 \tag{5}$$

Where the above square is elementwise. As an intuitive explanation, the second order interactions are modelled by several steps: i) embedding of the one-hot per-field feature using linear layer $E_F$. ii) projection of the feature embedding to a higher dimension using the per-field projection matrix $U_F$. iii) summation over the projected embeddings of all fields. iv) computing the scalar product of their square with the diagonal of the matrix $\Lambda$. Note that this can be expressed by a shallow neural network that first learns a per-field representation, then aggregates the representations by summation, passes the result through a non-linearity, and then a linear layer. Factorization machines are a particular instantiation of this neural network, when the activation function is a quadratic function. This will be generalized in the next section.

## 3 FIELDWISE FACTORIZED NETWORKS FOR TABULAR CLASSIFICATION

We propose *fieldwise-factorized neural-networks*, a general framework for tabular classification.

### 3.1 GENERAL FRAMEWORK

In this section, we overview our proposed framework: *fieldwise-factorized neural-networks* (F2NN), while detailing its components in the following sections. We learn a dedicated neural network $\phi_F$ for each field that takes as input the values of the field $\boldsymbol{x_F}$ (one-hot for categorical fields, scalar for numeric fields) and returns a high-dimensional embedding $\phi_F(\boldsymbol{x_F})$. The architecture of the fieldwise neural network will be described in Sec. 3.2. The aggregated high-dimensional feature is obtained by summing over the fields $\sum_F \phi_f(\boldsymbol{x_F})$. It is then passed through an activation function $\sigma$. Our activation $\sigma$ allows different activation functions for different dimensions. The post-activation results are finally multiplied by output linear layer $W_{out}$, mapping it to the output logits $z_{out}$:

$$z_{out} = W_{out}\sigma(\sum_F \phi_F(\boldsymbol{x_F})) \tag{6}$$

Note that differently from factorization machine, but similarly to other neural networks, our framework is able to handle both binary and multiclass classification tasks.

The framework is characterized by two main components: per-field representation by factorized networks $\phi_F$, and an adaptive activation $\sigma$ on the aggregated representations of all fields. We will describe them in detail in the next sections.

### 3.2 FIELDWISE FACTORIZED NETWORKS

The theoretical basis in Sec. 2 suggests that the learned field embeddings should be of high-dimension to factorize a full-rank $M_{F(i),F(j)}$, but that these representations should be low-rank factorized. The motivation behind these design choices is to reduce sample complexity. Modeling field interactions requires a high-dimension while there may not be a sufficient number of samples per-feature for estimating it. Complexity is reduced: i) by learning a per-field representation that does not take other features into account ii) by using low-rank factorization inside these fieldwise networks. Since we are able to use other activation than just than quadratic, $d$ is allowed to be even larger than the full rank dimension. Our framework is able to handle both categorical and numeric fields:

**Categorical fields:** Following Sec. 2.2, we learn fieldwise networks $\phi_F^{cat}(\boldsymbol{x_F}) = U_F E_F \boldsymbol{x_F}$. We choose the per-feature embedding (output of $E_F$) to be low-dimensional as there are often many features and limited data per-feature, while matrix $U_F$ projects this to high-dimension.

**Numeric fields:** Differently from categorical fields, numeric fields are ordered which enables learning more complex functions. We choose to learn a factorized one-hidden layer for each field. Our network first projects the scalar value to a high-dimension $\boldsymbol{t_F} = \boldsymbol{v_F} x_F + \boldsymbol{b_F}$ (where $\boldsymbol{v_t}, \boldsymbol{b_F} \in R^l$). The results are passed through a ReLU network, and mapped to the per-field embedding. We found that a high dimensional $\boldsymbol{t_F}$ is important for achieving strong performance. However, as $\boldsymbol{t_F}$ and the output field-embedding $\phi_F$ have a high-dimension, the second linear layer becomes very large. Instead, we choose to low-rank factorize the second layer. The post-activation $ReLU(\boldsymbol{t_F})$ is projected to a low-dimension, by linear layer $E_F$ and is then projected using linear layer $U_F$ to the high-dimensional field embedding $\phi_F$. Note that as both $E_F$ and $U_F$ are linear, with no intermediate non-linearity - they are equivalent to a single (low-rank) linear layer. The entire network is therefore a factorized one-hidden layer neural network.

In summary, the fieldwise factorized networks for categorical and numeric variables are given by:

$$\phi(\boldsymbol{x_F}) = \begin{cases} U_F E_F \boldsymbol{x_F} & Categorical \\ U_F E_F ReLU(\boldsymbol{v_F} x_F + \boldsymbol{b_F}) & Numeric \end{cases} \tag{7}$$

Note that although SAINT (Somepalli et al. (2021) also use a one-hidden layer network for numeric values, its formulation does not use the low-rank factorization and therefore cannot handle high width, which we show is key to the performance of our method. Also note that our formulation can easily handle deeper fieldwise architectures, but we did not observe benefits from deeper networks.

### 3.3 OUR FRAMEWORK CAN EXPRESS POPULAR TABULAR CLASSIFICATION METHODS

Despite the simplicity of our framework, it is very general. By changing the fieldwise factorized network $\phi_F$ and activation $\sigma$, it can express several popular tabular classification methods:

*Factorization machine methods:* When choosing a quadratic activation $\sigma(x) = x^2$ and the fieldwise network $\phi_F(\boldsymbol{x_F}) = U_F E_F \boldsymbol{x_F}$ our framework becomes a factorization machine-based model. This proof was detailed in Sec. 2.2. Therefore, factorization machine models are expressible by our framework.

*One-hidden Layer ReLU Networks:* When choosing the activation $\sigma(x) = ReLU(x)$ and the fieldwise networks as simple linear layers $\phi_F(x) = U_F x$, our framework becomes a one-hidden layer neural network. Although more layers can be easily added to our framework after the activation layer $\sigma$, we did not find this beneficial in practice. We will present empirical results in Sec. 4.2 for showing this.

*Wide-and-Deep:* When selecting a fraction of dimensions of $\sigma$ to have quadratic activations, while the rest are selected to have ReLU activation, our framework becomes a wide-and-deep model (as it is the sum of a ReLU network and a factorization machine). Although the original wide-and-deep method used full rank for the feature interaction $W_{ij}$, later methods (e.g. Guo et al. (2017)) use different factorization machine varieties. By modifying the choice of dimensionality of the per-feature embedding $\boldsymbol{e_i}$ and per-field projection $d$, all the above methods are expressible by our model. One caveat is that the deep part of our framework only has a single hidden layer, but as mentioned previously, adding further layers has not improved results in our experiments.

## 3.4 Adaptive Non-Linearity

In Sec. 3.3 we established that by choosing different activation function $\sigma$, our framework can express a range of popular tabular classification methods. We will show in Sec. 4.3, that in different classification problems, either $\sigma(x) = ReLU(x)$ or $\sigma(x) = x^2$ yields significantly better performance. By the analysis in Sec. 3.3 these are cases where either ReLU neural networks or factorization machine achieve better results. By selecting $\sigma$ a concatenation of ReLU and quadratic functions, we may be able to deal with all cases. However this may not be an efficient solution as it doubles the number of parameters. It would naturally be attractive to use a non-linearity that may be able to automatically adapt to the setting most beneficial to a particular dataset. Here we suggest using the GELU activation function (Hendrycks & Gimpel, 2016) as a bridge between the two paradigms. At the limits, $GELU(x)$ has the following attractive properties:

$$GELU(x) \approx \begin{cases} 0.5x + \sqrt{\frac{1}{2\pi}}x^2 & \|x\| << 1 \\ ReLU(x) & \|x\| \to \infty \end{cases} \tag{8}$$

Therefore, by varying the magnitude of the input, and utilizing the GELU activation, our framework is able to switch between factorization machine and ReLU network behavior. The adaptive behavior will be demonstrated empirically in Sec. 4.3. GELU can be used as an efficient alternative for exhaustive search over the optimal activation function or an ensemble approach (wide-and-deep).

## 3.5 Practical Considerations: The Devil is In The Details

The field of deep learning has made significant steps forward by increasing the depth of neural networks and by using attention-based architectures such as transformers. It might be expected that increasing network depth or using transformers might be beneficial for tabular data. Unfortunately, in our experiments, we did not find evidence to support this. We do not claim that it is impossible to gain by using the two mechanisms, but this is not trivial. More optimistically, in our experiments we found that there are lower hanging fruit. Specifically, we obtained significant gains by choosing suitable width $d, l$ and activation $\sigma$, as well as tuning the regularization strength (either dropout or weight decay). We suggest GELU and full rank projections $U_F$ ($d = J \cdot K$) as robust default values. Although tuning the embedding dimension $K$ can also help, we kept it in-line with the baselines.

## 4 EXPERIMENTS

### 4.1 PERFORMANCE COMPARISON

#### 4.1.1 GENERAL TABULAR CLASSIFICATION

**Evaluation protocol** We evaluated our method on the same datasets as SAINT [1] (Somepalli et al. (2021)). The datasets include 14 binary classification tasks and 2 multiclass classification tasks. We did not include Arcene and Arrythmia datasets from the reported results as they are very small datasets and their results have high variance (Arrhythmia contains 452 samples and Arcene contains 200 samples). In addition, we use exactly the same pre-processing as in the official SAINT implementation. Details of these datasets and the pre-processing are provided in Appendix C.

**Hyperparameters**: Our optimization hyper-parameters follow SAINT. We run a grid search on the hyperparameters presented in Table 5 with seed=1, and selected the best model by the validation set. For fair comparison, we used the same $E_F$ dimensionality as SAINT defaults.

**Baselines** We compare against the baselines reported in SAINT. They include traditional methods such as logistic regression and random forests, as well as powerful boosting libraries XGBoost (Chen & Guestrin (2016)), LightGBM (Ke et al. (2017)), and CatBoost (Prokhorenkova et al. (2018)). They also compare against deep learning methods, like multi-layer perceptrons, VIME (Yoon et al. (2020)), TabNet (Arik & Pfister (2021)), and TabTransformer (Huang et al. (2020)). The numbers were copied from the SAINT paper. In order to compare the performance on the same exact seeds, we re-run the SAINT models on 10 different seeds (1-10). For a fair comparison, we trained SAINT with our best numerical embedding hidden dimension $l$ and their default selection (100), and reported the best of the two settings.

**Results** The averaged performance for all methods is reported in Table 1, the results per dataset are detailed in Table 9. We reported ROC-AUC for binary classification, and class prediction accuracy or multi-class classification. Our results are slightly better or comparable to SAINT variations, except from on multiclass tasks, where SAINT-s achieves poor performance. Our method also outperforms all of SAINT's baselines on average. SAINT-s and our framework use exactly the same learning setting (optimization hyper-parameters, datasets). The difference between the architectures is the post-$E_F$ architecture. While we use a set of linear projections that are only summed and passed through activation and classification layers, SAINT-s uses a massive six layer Transformer. This highlights that for robust and accurate neural models for tabular data, it does not appear that complex methods have an edge over well-tuned simple models such as ours.

#### 4.1.2 CTR PREDICTION

CTR prediction is a fundamental tabular binary classification task. FM-based models are the most common approach for tackling it. As our framework generalizes FM-based models, we also examined if CTR prediction can benefit from our framework.

**Datasets** We used Avazu and Criteo which are common CTR prediction benchmarks.

**Baselines** We compared the framework implementation against FmFM, which is the shallow state-of-the-art of FM-models, and two deep variations - DeepFwFM and DeepFmFM. We implemented our framework on the FmFM public code-base [2], and used their data processing and baselines implementation. We split the data as described in FmFM paper, and applied the same splits for all the models. We conducted a hyper-parameter search for both our method and FmFM, and found the optimal hyperparameters for both were the same. We thus applied the same hyperparameters for the deep baselines. Other implementation details are provided in Appendix C.

**Results** The ROC-AUC results are reported in Table 1. We reported the results of using the papers hyperparameters and using our tuned hyperparameters, as well as quoted the reported results from the paper. We can observe that even after improving the baselines by a better tuning, our framework outperforms all other baselines.

---

[1] https://github.com/somepago/saint
[2] https://github.com/yahoo/FmFM

Table 1: Performance comparison. **Left:** general tabular data classification (Average over all datasets). Above line - results taken from SAINT paper evaluated on unknown seeds. Below line - mean and standard deviation of SAINT results computed using the authors' code averaged over seeds 1-10, and of our method evaluated on exactly the same data. **Right:** CTR prediction (ROC-AUC %). **I** results reported by authors, **II** results of running the model on our data split with paper's hyperparameters, **III** result of running the model on our data split with our tuning for the model. DeepFwFM results are without a pruning mechanism.

|                     | **All**            |
|---------------------|--------------------|
| Logistic Regression | 86.822             |
| Random Forest       | 89.086             |
| XGBoost             | 91.038             |
| LightGBM            | 89.658             |
| CatBoost            | 90.197             |
| MLP                 | 84.106             |
| VIME                | 78.482             |
| TabNet              | 86.515             |
| Tab Transformer     | 88.654             |
| SAINT               | 91.884             |
| SAINT-i             | 91.716             |
| SAINT-s             | 90.669             |
| SAINT               | 91.596 $\pm$0.108  |
| SAINT-i             | 91.286 $\pm$0.142  |
| SAINT-s             | 89.821 $\pm$0.230  |
| Ours                | 91.635 $\pm$0.158  |

| Model    | Criteo |        |        |
|          | I      | II     | III    |
|----------|--------|--------|--------|
| FmFM     | 81.09  | 81.11  | 81.166 |
| DeepFmFM | -      | -      | 81.175 |
| DeepFwFM | 81.16  | 81.105 | 81.130 |
| Ours     | -      | -      | 81.245 |

| Model    | Avazu  |        |        |
|          | I      | II     | III    |
|----------|--------|--------|--------|
| FmFM     | 77.63  | 77.343 | 78.635 |
| DeepFmFM | -      | -      | 78.647 |
| DeepFwFM | 78.93  | 78.519 | 78.596 |
| Ours     | -      | -      | 78.701 |

## 4.2 ABLATION STUDIES

In this section we will explore the effect of the framework components on its performance. The experiments in the following section were conducted using SAINT evaluation datasets and process. From resources constraints, we run the experiments on seed=1 only. The results are reported in Table 2, averaged over the datasets, more details are found in Tables 10-11.

**Per-field Projections:** To examine the importance of the per-field networks, we evaluated sharing the projection matrix $U_F$ across fields. We can observe that all datasets (besides HTRU2), gain from the fieldwise projections. In multi-class datasets, the difference is much significant.

**Low rank projections:** In this experiment we reduced the dimension of the projection matrix $U_F$ to that of the output of $E_F$. Overall, we can observe, that the factorized high-dimensional representation is crucial, especially in Forest, Volkert and MNIST.

**Numeric fieldwise networks:** We tested the following alternatives: (i) a $K$-dimensional linear layer for each field, (ii) applying non-linearity, without a high-dimensional projection, meaning $v_F$ dimension is equal to $E_F$ dimension, (iii) projecting to high-dimensional space and reduce to $K$-dimensional, without a non-linearity. The averaged results over the datasets that contain continuous fields are reported in Table 2 (Right). We can observe that the non-linearity is an important aspect, and boost the performance. In addition, by controlling the width for the high-dimensional space, we gain a further improvement over the low-dimensional non-linear network.

**Deeper model:** We added 2 extra layers after the activation $\sigma$ with equal width as field embedding, $\phi(x_F)$, and matching the model activation and dropout. Since quadratic activation performs poorly when deepening the network, we tested all activations for the datasets that selected a quadratic in their shallow variation, and report the best. While we observed improvement on specific datasets, such as Volkert and Forest, in most of the datasets the results were comparable of even worse than our shallow model with the linear classification head.

**The activation after aggregating the fieldwise representations** In the original selected models, the selected activations are divided as follow: 6 datasets with a quadratic activation, 4 datasets with ReLU activation, and 4 with GELU activation. In order to explore the effect of the activation, we vary the activation for each dataset, while keeping the other hyperparameters fixed. The averaged

Table 2: Averaged ablation studies. **Left:** General framework architecture ablations. **Right:** Numeric fieldwise networks ablations.

| Variation | Binary (ROC-AUC) | Multiclass (Accuracy) | All |
|---|---|---|---|
| Full method | 93.16 | 84.02 | 91.85 |
| Shared field matrix | 92.55 | 70.47 | 89.40 |
| Deep | 92.89 | 82.57 | 91.41 |
| Low rank | 92.64 | 78.97 | 90.69 |
| Quadratic | 92.70 | 82.31 | 91.21 |
| ReLU | 93.05 | 82.58 | 91.55 |
| GELU | 93.00 | 82.61 | 91.51 |

| Variation | Num. DS |
|---|---|
| Full method | 91.40 |
| (i) $U_F E_F x_F$ | 90.74 |
| (ii) $l = K$ | 91.17 |
| (iii) $U_F E_F (\boldsymbol{v_F} x_F + \boldsymbol{b_F})$ | 90.80 |

Table 3: GELU as adapted activation - HR@10 on Movielens dataset

| Emb. size | ReLU | GELU | Quad. | MF |
|---|---|---|---|---|
| 16 | 0.6876 | 0.6972 | 0.6974 | 0.6937 |
| 192 | 0.7164 | 0.7217 | 0.7280 | 0.7278 |

results are reported in Table 2. In many datasets the particular activation was immaterial. However, in some datasets, activation has made a significant difference, so it should be carefully chosen.

### 4.3 GELU AS AN AUTOMATICALLY ADAPTED ACTIVATION

We conducted an empirical study on the advantage of using GELU activation, by examining two cases - a case where quadratic activations are preferred over ReLU, and a case where ReLU activations are preferred. We demonstrate that by using GELU, we can get the closest result to the preferred activation.

*Quadratic is better than ReLU:* Rendle (2010) showed that a simple matrix factorization model performs better than MLP and NeuMF models (Zhao et al. (2021)) for collaborative filtering. We applied our method on the authors' dataset and settings. We tested the effect of different choices of $\sigma$ on the ability of our framework to express a dot product between fields. Dedicated choices of initialization and regularization were used, see details in Appendix C. We report the results for embedding sizes $K = 16$ and $K = 192$, the minimal and maximal embedding sizes that were tested in Rendle (2010). Note that the reported result for matrix factorization of Rendle (2010) is $0.7294$ (on embedding size of 192). As expected, we can observe that for each embedding size, quadratic activation performs the best, while ReLU performs poorly. However, GELU achieves similar results to the quadratic performance, demonstrating its robustness and adaptive behaviour.

*ReLU is better than Quadratic:* On the other hand, from our experiments on forest dataset, we observed that a quadratic activation performs dramatically worse than ReLU ( $94.84\%$ and $99.4\%$, respectively). However, GELU obtains $99.47\%$ which is in line with ReLU.

### 5 DISCUSSION AND CONCLUSIONS

In this paper, we presented a theoretically grounded, general framework for handling tabular data classification. Our framework can be extended in many ways. Two promising ideas are custom activations and representation transfer:

*Activation for tabular tasks.* We highlighted the importance of the activation $\sigma$ and demonstrated both theoretically and empirically, that GELU activation is suitable across the full range of tabular tasks. This suggests that developing an activation function for tabular classification is very promising.

*Exploring the learned representations.* As a by-product of tabular data classification, our framework learns fieldwise representations. Future work should examine if the representations are useful for transfer learning for related tasks.

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

## A    APPENDIX - F2NN FRAMEWORK ILLUSTRATION

An illustration of our framework can be found in Figure 5.

## B    APPENDIX - FM-BASED MODELS FORMULATIONS

**Embedding form:** FM, FwFM and FmFM, learn $C$ vectors, one for each feature, $e_i \in \mathbb{R}^K$.

FFM learns $(J-1)$ vectors for each feature, $e_i^{F(j)} \in R^{\frac{K}{J}}$, overall $C \cdot (J-1)$ vectors. Note that they select much higher $K$ than all other approaches, therefore $\frac{K}{J} > 1$. $e_i \in R^K$ places these $(J-1)$ vectors in the relevant indices, while $0_{\frac{K}{J}}$ in its field indices. For example, $e_1 = [0_{\frac{K}{J}}, e_1^2, ..., e_1^J]$, $e_2 = [e_2^1, 0_K, ..., e_2^J]$. Then, $P_F(j) \in \mathbb{R}^{K \times K}$ ($\mathbb{R}^{J \cdot \frac{K}{J} \times K}$) extracts the embedding $e_i^{F(j)}$. For example, $P_2$ is of the form: $P_2 = \begin{bmatrix} 0_{\frac{K}{J}} \\ I_{\frac{K}{J}} \\ 0_{\frac{K}{J}} \\ . \\ . \\ . \\ 0_{\frac{K}{J}} \end{bmatrix}$

**Interaction matrix form:** In general, for every FM-based model, M takes the following form:

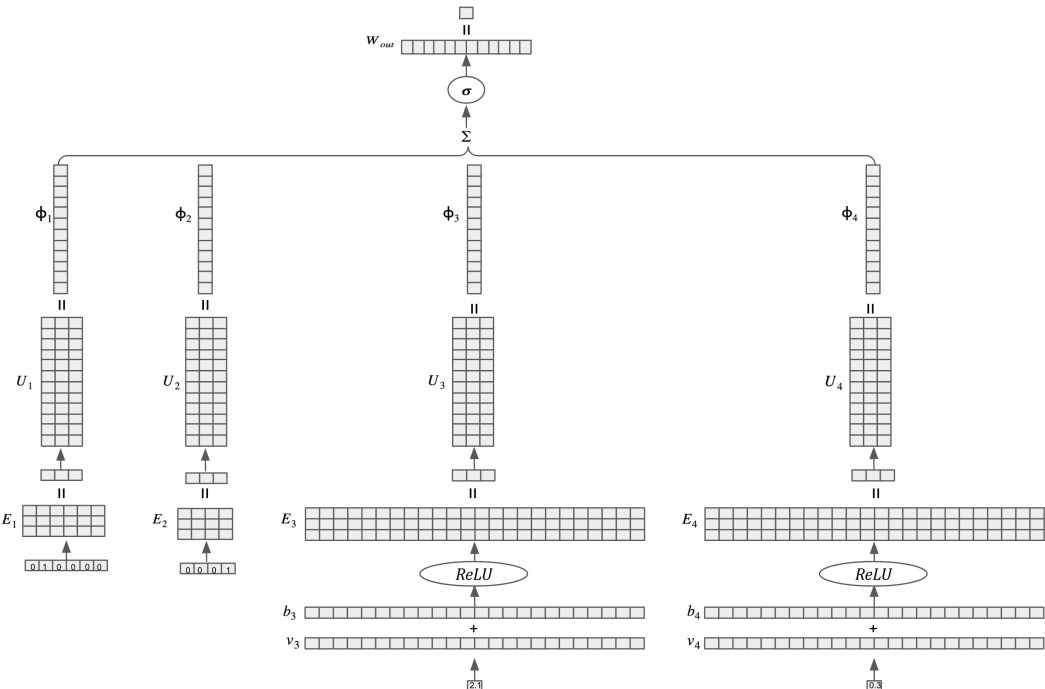

Figure 1: Fieldwise factorized neural networks (F2NN) architecture. Toy example of binary classification: Fields 3 and 4 are numerical, while fields 1 and 2 are categorical, with six and four categories, respectively.

$$M = \frac{1}{2} \cdot \begin{bmatrix} 0_K & M_{1,2} & M_{1,3} & ... & M_{1,J} \\ M_{1,2} & 0_K & M_{2,3} & ... & M_{2,J} \\ . & . & . & ... & . \\ . & . & . & ... & . \\ . & . & . & ... & . \\ M_{1,J} & M_{2,J} & M_{3,J} & ... & 0_K \end{bmatrix} \tag{9}$$

# C  APPENDIX - EXPERIMENTAL SETTINGS AND IMPLEMENTATION DETAILS

## C.1  PERFORMANCE EVALUATION

### C.1.1  GENERAL TABULAR DATA

**Datasets** Detailed information on the evaluated datasets is reported in table 4. It can be seen that the evaluated datasets are diverse, in terms of their size and the amount of fields, and contains both categorical and numerical features. Each of these datasets is publicly available from either UCI [3] or AutoML [4]. We use the exact processing as is SAINT implementation, i.e. all the continuous features are Z-normalized, and all categorical features are label-encoded before the data is passed on to the embedding layer.

**Implemetation details** We implemented our framework in the SAINT public code-base [5]. Linear and Embedding layers were initialized using the PyTorch default. We used dropout before $U_F$ and after the activation, and a bias for $U_F$ and for the numerical $E_F$ and $t_F$. As in factorization

---

[3] http://archive.ics.uci.edu/ml/datasets.php
[4] https://automl.chalearn.org/data
[5] https://github.com/somepago/saint

Table 4: General tabular datasets

| Dataset | Task | #Fields | #Categ. | #Numer. | Size | #Positives | #Negatives | % of positives |
|---|---|---|---|---|---|---|---|---|
| Income | Binary | 14 | 8 | 6 | 32,561 | 7,841 | 24,720 | 24.08 |
| Bank | Binary | 16 | 9 | 7 | 45,211 | 5,289 | 39,922 | 11.7 |
| BlastChar | Binary | 20 | 17 | 3 | 7,043 | 1,869 | 5,174 | 26.54 |
| Credit | Binary | 29 | 0 | 29 | 284,807 | 492 | 284,315 | 0.17 |
| Forest | Binary | 49 | 0 | 49 | 495,141 | 283,301 | 211,840 | 57.22 |
| HTRU2 | Binary | 8 | 0 | 8 | 17,898 | 1,639 | 16,259 | 9.16 |
| KDD99 | Binary | 39 | 3 | 36 | 494,021 | 97,278 | 396,743 | 19.69 |
| Shoppers | Binary | 17 | 2 | 15 | 12,330 | 1,908 | 10,422 | 15.47 |
| Philippine | Binary | 308 | 0 | 308 | 5,832 | 2,916 | 2,916 | 50 |
| QSAR Bio | Binary | 41 | 0 | 41 | 1,055 | 356 | 699 | 33.74 |
| Shrutime | Binary | 11 | 3 | 8 | 10,000 | 2,037 | 7,963 | 20.37 |
| Spambase | Binary | 57 | 0 | 57 | 4,601 | 1,813 | 2,788 | 39.4 |
| Volkert | Multiclass(10) | 147 | 0 | 147 | 58,310 | - | - | - |
| MNIST | Multiclass(10) | 784 | 784 | 0 | 60,000 | - | - | - |

Table 5: Saint datasets - Architecture hyperparameters search. $d_{factor}$ and $l_{factor}$ stands for factors of the full rank dimensionality of $E_F$ and $t_F$ respectively.

| Parameter | Values |
|---|---|
| $d_{factor}$ | $\{0.1, 0.25, 0.5, 0.75, 1, 2, 3, 4, 5\}$ |
| $l_{factor}$ | $\{1, 2, 3, 4\}$ |
| Activation | $\{ReLU, Quadratic, GELU\}$ |
| Dropout | $\{0, 0.1, 0.25, 0.5, 0.75\}$ |

machine methods, we observe minor benefits from adding linear per-field terms to the final logits. We implemented this by adding to the per-field final layer $W_F$, an identity matrix with the same dimension as the output of $E_F$. The activation $sigma$ for the resulting dimension was chosen to be identity $\sigma(x) = x$.

**Training** Our optimization hyper-parameters follow SAINT. We used the AdamW optimizer with $\beta_1 = 0.9$, $\beta_2 = 0.999$, decay = 0.01, and with a learning rate of $0.0001$ with batches of size 256 (except for MNIST, which they use a batch size of 32). We trained for 100 epochs. We split the data into 65%, 15%, and 25% for training, validation, and test splits, respectively.

**Hyperpararmter search:** We performed architecture search on the hyperparameters presented in Table 5 with seed=1. We selected the best model by the validation set. Our experiments employed a fixed embedding and projection sizes to all of the fields (although tuning this might improve results).

### C.1.2 CTR PREDICTION

**Datasets:** The first one is the Criteo CTR data set, it is a well-known benchmark dataset which used for the Criteo Display Advertising Challenge. There are 45 million samples and each sample has 13 numerical fields and 26 categorical fields. The second data set is the Avazu CTR data set, it was used in the Avazu CTR prediction competition, which predicts whether a mobile ad will be clicked. There are 40 million samples and each sample has 23 categorical fields.

**Implementation details** We implemented our framework on FmFM public code-base [6], and used their data processing and baselines implementation. We split the data as described in FmFM paper, and applied the same splits for all the models. For our implementation, we adopted FmFM model implementation, and changed the main component to a Tensorflow linear layers, an activation between them, and bias for the per-field projection layer. For the weights initialization, FmFM implementation uses weights that are sampled from a normal distribution. For the reported frame-

---

[6]https://github.com/yahoo/FmFM

Table 6: CTR prediction baselines hyperparameters: learning rate($\gamma$), regularization strength ($\lambda$), standard deviation of the weights normal initialization ($\sigma$), batch size ($bs$). **I** results reported by authors, **II** results of running the model on our data split with paper's hyperparameters, **III** result of running the model on our data split with our tuning for the model.

| Model | Criteo | | |
|---|---|---|---|
| | **I** | **II** | **III** |
| FmFM | $\gamma = 1e^{-4}, \lambda = 1e^{-5},$ $\sigma = 0.01, bs = 1024$ | $\gamma = 1e^{-4}, \lambda = 1e^{-5},$ $\sigma = 0.01, bs = 1024$ | $\gamma = 1e^{-4}, \lambda = 1e^{-5},$ $\sigma = 0.2, bs = 2000$ |
| DeepFmFM | - | - | $\gamma = 1e^{-4}, \lambda = 1e^{-5},$ $\sigma = 0.2, bs = 2000$ |
| DeepFwFM | $\gamma = 1e^{-3}, \lambda = 3e^{-7},$ $\sigma = 0.01, bs = 2048$ | $\gamma = 1e^{-4}, \lambda = 1e^{-5},$ $\sigma = 0.01, bs = 2048$ | $\gamma = 1e^{-4}, \lambda = 1e^{-5},$ $\sigma = 0.2, bs = 2000$ |
| Ours | - | - | $\gamma = 1e^{-4}, \lambda = 1e^{-5},$ $\sigma = 0.2, bs = 2000$ |

| Model | Avazu | | |
|---|---|---|---|
| | **I** | **II** | **III** |
| FmFM | $\gamma = 1e^{-4}, \lambda = 1e^{-5},$ $\sigma = 0.01, bs = 1024$ | $\gamma = 1e^{-4}, \lambda = 1e^{-5},$ $\sigma = 0.01, bs = 1024$ | $\gamma = 1e^{-3}, \lambda = 2e^{-6},$ $\sigma = 0.2, bs = 5000$ |
| DeepFmFM | - | - | $\gamma = 1e^{-3}, \lambda = 2e^{-6},$ $\sigma = 0.2, bs = 5000$ |
| DeepFwFM | $\gamma = 1e^{-3}, \lambda = 6e^{-7},$ $\sigma = 0.01, bs = 2048$ | $\gamma = 1e^{-4}, \lambda = 1e^{-5},$ $\sigma = 0.01, bs = 2048$ | $\gamma = 1e^{-3}, \lambda = 2e^{-6},$ $\sigma = 0.2, bs = 5000$ |
| Ours | - | - | $\gamma = 1e^{-3}, \lambda = 2e^{-6},$ $\sigma = 0.2, bs = 5000$ |

work instance performance, in Criteo we used ReLU activation and a full rank per-field projection layer, and for Avazu we used GELU activation and factor of 5 of the full rank dimension. In this settings, we did not use a dropout between the layers. Note that since the numerical fields in Criteo datasets are discrete, we used the same numerical data processing as in the baselines and consider them as categorical.

**Hyperparameters search** We found that the hyperparameters in this task are crucial, included the standard deviation of the weights initialization. Therefore, we performed an hyperparameters search both for our framework and for FmFM, though obtained that they should use the same hyperparameters. Thus, we applied this hyperparameters for the deep baselines too. For the hyperparameters search, we implemented our framework by a GELU activation and a full rank per-field projection layer. The exact hyperparameters of each baseline is reported in Table 6.

### C.1.3 ACTIVATION STUDY

Rendle (2010) public code is optimized for simplicity and not for efficiency, therefore we have implemented an efficient keras version of it, using Dot layer, and adjusted the hyperparamters to fit their reported results. We used Adam optimizer, batch size of $10k$ and L2 regularization strength of $2.5e^{-7}$, while kept the negative sampling on 8, and a learning rate of $0.002$.

Next, since matrix factorization is a special case of our framework, we implemented it by our framework, using quadratic activation and the eigen vectors and values of $0.5 \cdot \begin{bmatrix} 0_K & I_K \\ I_K & 0_K \end{bmatrix}$ for the per-field projections and for the final classification layer, respectively. The eigenvectors were partitioned with respect to the fields. When freezing these weights, we get an exact implementation of a dot product, which also reflected by a replication of the performance between both of the implementations.

Table 7: GELU as adapted activation - regularization strength of the reported results

| | MF initialization & regularization | | |
|---|---|---|---|
| **Emb. size** | **ReLU** | **GELU** | **Quad.** |
| 16 | $\lambda = 1e^{-5}$ | $\lambda = 1e^{-5}$ | $\lambda = 1$ |
| 192 | $\lambda = 5e^{-3}$ | $\lambda = 5e^{-4}$ | $\lambda = 1$ |

In order to explore the ability of other activation to approximate the results of a dot product based model, we used the eigen decomposition as an initialization and regularization, and searched the best regularization strength for each activation. The regularization strength of the reported results are reported in Table 7.

We reported this experiments results on embedding sizes of 16 and 192, the minimal and maximal embedding sizes that were tested in Rendle (2010).

## D   APPENDIX - RESULTS PER DATASET

**Performance comparison:** The performance comparison per dataset are reported in Table 9. The results are averaged over seeds 1-10. We reported also the standard deviation.

**Ablation studies:** Tables 10-11 report the results of the ablation studies per dataset

## E   APPENDIX - INFERENCE TIME COMPARISON

We also examined the runtime between our framework and SAINT. We computed the average inference time over all batches of size 256. The results on three datasets are reported in Table 8. We can observe that using a transformer slows inference without an improvement to the performance.

Table 8: Inference time comparison (milliseconds per batch on average). F2NN is faster than SAINT variations.

| | Income | Forest | Bank |
|---|---|---|---|
| SAINT-s | 6.890 | 7.621 | 5.764 |
| SAINT-i | 1.502 | 3.327 | 1.348 |
| SAINT | 2.449 | 4.480 | 2.107 |
| F2NN | 0.660 | 0.933 | 0.575 |

Table 9: Performance comparison per dataset, Averaged results over seeds 1-10.

| | ROC-AUC | | | | | | |
| --- | --- | --- | --- | --- | --- | --- | --- |
| | Credit | HTRU2 | QSAR Bio | Spambase | Shrutime | Philippine | KDD99 |
| SAINT | 97.89 ±0.85 | 97.98 ±0.42 | 93.28 ±1.4 | 98.37 ±0.44 | 86.37 ±1.07 | 81.22 ±1 | 100 ±0 |
| SAINT-i | 97.92 ±0.85 | 98.02 ±0.35 | 92.9 ±1.79 | 98.2 ±0.34 | 86.23 ±1.23 | 80.56 ±1.48 | 100 ±0 |
| SAINT-s | 98.09 ±0.72 | 97.99 ±0.44 | 92.64 ±1.41 | 98.28 ±0.21 | 86.18 ±1.11 | 78.46 ±1.81 | 100 ±0 |
| F2NN | 97.01 ±1.31 | 98.13 ±0.39 | 92.89 ±1.48 | 98.31 ±0.39 | 86.47 ±1.15 | 80.81 ±1.88 | 100 ±0 |

| | ROC-AUC | | | | | Accuracy | |
| --- | --- | --- | --- | --- | --- | --- | --- |
| | Bank | Blastchar | Forest | Shoppers | Income | Volkert | MNIST |
| SAINT | 92.93 ±0.4 | 84.03 ±0.93 | 98.2 ±0.24 | 93.2 ±0.47 | 91.44 ±0.36 | 69.57 ±0.4544 | 97.86 ±0.1035 |
| SAINT-i | 93.12 ±0.4 | 83.98 ±0.9 | 94.74 ±0.97 | 92.89 ±0.41 | 91.4 ±0.38 | 70.34 ±0.2749 | 97.71 ±0.1774 |
| SAINT-s | 93.67 ±0.25 | 83.99 ±0.98 | 99.71 ±0.02 | 93.06 ±0.51 | 91.56 ±0.3 | 49.71 ±3.156 | 94.16 ±0.3232 |
| F2NN | 93.43 ±0.27 | 84.09 ±0.94 | 99.49 ±0.02 | 92.97 ±0.48 | 91.54 ±0.36 | 70.22 ±0.3784 | 97.53 ±0.1266 |

Table 10: Ablations per dataset

| | ROC-AUC | | | | | | | | | | | | Accuracy | |
| --- | --- | --- | --- | --- | --- | --- | --- | --- | --- | --- | --- | --- | --- | --- |
| | Credit | HTRU2 | QSAR Bio | Shrutime | Spambase | Philippine | KDD99 | Bank | Blastchar | Forest | Shoppers | Income | Volkert | MNIST |
| F2NN | 97.73 | 98.42 | 94.12 | 86.6 | 98.44 | 82.47 | 100 | 93.09 | 82.38 | 99.47 | 93.53 | 91.62 | 70.35 | 97.7 |
| Low-rank | 96.95 | 98.57 | 93.49 | 86.69 | 98.09 | 81.41 | 100 | 92.88 | 82.41 | 96.35 | 93.2 | 91.67 | 65.27 | 92.66 |
| Shared | 95.93 | 98.42 | 93.48 | 85.51 | 98.02 | 81.02 | 100 | 92.09 | 82.3 | 99.42 | 92.96 | 91.5 | 62.44 | 78.5 |
| Deep | 97.48 | 98.25 | 93.54 | 85.39 | 98.24 | 81.47 | 100 | 93.11 | 82.3 | 99.58 | 93.54 | 91.77 | 71.9 | 93.24 |
| Quadratic | 97.73 | 98.47 | 93.81 | 86.6 | 98.11 | 82.47 | 100 | 92.95 | 82.38 | 94.84 | 93.4 | 91.62 | 66.93 | 97.7 |
| ReLU | 97.89 | 98.42 | 94.12 | 85.99 | 98.44 | 81.76 | 100 | 93.06 | 82.25 | 99.4 | 93.53 | 91.74 | 69.93 | 95.22 |
| GELU | 97.71 | 98.42 | 93.77 | 85.93 | 98.4 | 81.73 | 100 | 93.09 | 82.26 | 99.47 | 93.49 | 91.69 | 70.35 | 94.88 |

Table 11: Numerical networks ablastions per dataset. (i) stands for $U_F E_F x_F$, (ii) represents $l = K$, and (iii) represents $U_F E_F (v_F x_F + b_F)$

| | ROC-AUC | | | | | | | | | | | | Accuracy |
| --- | --- | --- | --- | --- | --- | --- | --- | --- | --- | --- | --- | --- | --- |
| | Credit | HTRU2 | QSAR Bio | Shrutime | Spambase | Philippine | KDD99 | Bank | Blastchar | Forest | Shoppers | Income | Volkert |
| F2NN | 97.73 | 98.42 | 94.12 | 86.6 | 98.44 | 82.47 | 100 | 93.09 | 82.38 | 99.47 | 93.53 | 91.62 | 70.35 |
| (i) | 97.84 | 98.48 | 92.11 | 85.25 | 97.78 | 79.55 | 100 | 92.86 | 81.89 | 99.13 | 92.76 | 91.19 | 70.78 |
| (ii) | 97.88 | 98.55 | 92.89 | 86.35 | 97.96 | 81.61 | 100 | 93.02 | 82.48 | 99.39 | 93.12 | 91.4 | 70.55 |
| (iii) | 96.04 | 98.3 | 94.58 | 85.49 | 98.01 | 79.44 | 100 | 92.72 | 81.96 | 99.1 | 93.18 | 91.27 | 70.26 |

