# OpenReview forum: "Fieldwise Factorized Networks for Tabular Data Classification"
_ICLR.cc/2022/Conference — ICLR 2022 Submitted_

### Official Review · Reviewer_J2FM · 2021-10-29

**Correctness:** 3
**Technical Novelty And Significance:** 3
**Empirical Novelty And Significance:** 3
**Recommendation:** 6
**Confidence:** 3

**Main Review:**

The authors present an interesting framework useful for tabular classification task.
They have compared their method with other classifiers in a fair way, with the experiments that are well designed and well described with the right amount of details useful for the reproducibility (I hope the authors share their code).
Moreover, the ablation studies are very interesting and seem to confirm the importance of the per-field network.

The results of the proposed approach are quite good even though it is difficult to be sure on which model is better because the results are really near each other and there isn't a difference statistically significant.
Nonetheless, the experiments have shown some interesting insights and the importance of some design choices.

My main concern is about the formalism they have used in the description of the method. I think that a revision of the math and formalism could make the work more readable and reproducible.
In the following some details.

The formalism introduced in Section 2 is not completely coherent and in some parts misleading.
There is not a different text formatting used for the scalar values and vectors (for instance bold for vectors). This good practice is instead followed in some parts of Section 3.
Just for example (but all the section 2 and 3 should be revised carefully in order to make the paper more rigorous):
- $x_f$ is a vector and $x_i$ is a scalar.
- $f$ is used to indicate the categorical features but also to indicates the generic field (in definition of $S_f$ for example). Probably it is better to use F to indicate the generic field.

Other comments:
- In §2.2 insert a reference to the Appendix where $M$ is appropriately described.
- In Equation 2 the authors introduce also $x_i$ and $x_j$ even though the previous defined factorization don't mention them. The authors should describe the motivation and the validity of this operation.

Some minor typos:
- Page 7: XGBoost Chen & Guestrin (2016) --> XGBoost (Chen & Guestrin (2016))
- Page 8: immaterial --> do you mean not significant?

Some doubts:
- $E_f$ is the field embedding matrix, and the embeddings are the rows (as described in §2.1). Before the Equation 4 authors stated that: $E_fx_f$ is the embedding of the one hot vector $x_f$ but, if $x_f$ is column vector, it should be $E_f^T x_f$. If I'm right the authors should carefully revised all equations accordingly.
- Equation 1 presents four Factorizazion Machine Models. If I understood well, only FmFM has been tested (probably is the best model among others). If it is correct the author should be clarify this point. Moreover, in Table 1 it is used the name FmFm but in other parts it is indicated as FmFM (also §4.1.2).

Missed references:
- [1] Yuanfei Luo et al. Network On Network for Tabular Data Classification in Real-world Applications 2020
- [2] Zhibin Li et al., Field-wise Learning for Multi-field Categorical Data, 2020

The [1] and [2] present two approaches to tackle the same problem. Moreover [2] seems to use a similar approach using a field-wise network. Could the authors describe the differences? Moreover, the models in [1] and [2] are not tested, why didn't they tested also these two methods? Do the other tested methods outperform them?

**Summary Of The Paper:**

The paper describes an approach for tabular data classification.
In particular, authors proposed the Fieldwise Factorized neural network to tackle the problem defining a general framework in which it is possible to express other tabular classification methods.
The proposed approach permits to manage categorical and numerical fields and seems to outperform other methods (on average). Moreover the experiments seem to demonstrate the importance of per-field network and the improvement in the results using GELU instead of the ReLU of Quadratic activations.

**Summary Of The Review:**

There are some interesting aspects and the insights are supported by a good experimental set. In my opinion, with a careful revision of the formalism in the description of the method, the paper can be accepted for ICLR.

---

> ### Author Response · Authors · 2021-11-22
> **Response to Reviewer J2FM**
>
> We thank the reviewer for the dedicated and positive review. We are pleased the reviewer recognized our framework and insights as interesting and highlighted our well-designed experimental set. The reviewer raised several valid points, which we believe can be addressed by the following clarifications and corrections.
>
> **Formatting of the mathematical parts:**
>
> ***“There is not a different text formatting used for the scalar values and vectors …. probably it is better to use F to indicate the generic field.”***
> Thank you, we changed the notation according to your suggestion. F now represents a general field. In addition, we use a bold notation for vectors to differentiate between scalars and vectors.
>
> ***“In Equation 2 the authors introduce also xi  and xj even though the previous defined factorization don't mention them. The authors should describe the motivation and the validity of this operation.”***: Thank you for pointing this out. $x_i$ and $x_j$ are the scalar indicators that were presented in section 2.1. We fixed $q_2$ definition (in the beginning of section 2.2) and added $x_ix_j$ to it.
>
> ***“...authors stated that: Efxf  is the embedding of the one hot vector xf but, if xf  is column vector, it should be EfTxf.”***: we fixed $U_F$ and $E_F$ dimensions: $U_{F} \in R ^{d \times K }$ , $E_F \in R ^ {K \times J_{F(i)}}$
>
> **CTR prediction - Performance comparison clarification**
>
> ***“If I understood well, only FmFM has been tested (probably is the best model among others). If it is correct the author should clarify this point.”***: Indeed, FmFM is the best model among the shallow FM-based versions, therefore we tested against it. However, we also compared to two deep versions of FM-based models: DeepFwFM (which is a little better than shallow FwFM) and Deep FmFM.
>
> **Missed references**
>
>  Thanks for sharing these works, they have been added to our related works.
>
> ***“Could the authors describe the differences?”***
>
> *[1]: NON*: Our fieldwise networks are factorized, meaning they rely on low-rank factorization and therefore enable high-dimensional representations, which are important according to the theoretical analysis.
> In addition, we highlight the importance of choosing a suitable non-linearity, while NON utilize only ReLU activation.
> Lastly, our components are much simpler yet effective. We aggregate the field representations by summation followed by a suitable non-linearity and simple linear layer. NON ensembles multiple aggregation mechanisms, such as attention and deep neural networks,  and apply further deep layers to fuse outputs of different operations. In our empirical experiments we demonstrated no gain from deepening our network, as well as competitive or slightly better performance over complex attention-based models.Therefore, it demonstrates that making these more complex is not necessarily needed, and highlights the importance of selecting suitable activations and widths in our simple architecture.
>
> *[2]: Field-wise Learning for Multi-field Categorical Data*:  Although they also utilize a low-rank factorization approach, our methods differ in multiple crucial aspects.
> First, they utilize linear models, while we showed the importance of selecting a suitable non-linearity. Our non-linear approach enables the model to express complex functions, while on the other hand, to model multiplication between the features.
> Second, in their model, the input for each fieldwise matrix is the other fields’ values, while our fieldwise networks consider only the specific field input. This means that their input is of much higher dimension, resulting in higher sample complexity and requiring more aggressive compression.
> In addition, they do not handle numerical fields (without discretizing them and converting them to categorical fields).
> Lastly, they evaluated their method only on CTR prediction task, while we also consider general tabular classification.
>
>
> ***"The models in [1] and [2] are not tested, why didn't they tested also these two methods? Do the other tested methods outperform them?"***: Both of these works evaluated their performance on CTR prediction or a recommendation system tasks. In our comparison, we focused on the latest and leading works of this domain, which are also a natural comparison as our framework generalizes them. The reported numbers from the papers are not comparable, they work on different split and preprocessing. However, since FWL published a public code, we would be able to include them in our comparison to the camera ready version.
>
> **Editorial comments:**
>
> *"In Table 1 it is used the name FmFm but in other parts it is indicated as FmFM.”*: Thank you for noticing, we fixed it.
>
> *“XGBoost Chen & Guestrin (2016) --> XGBoost (Chen & Guestrin (2016))”*: Thank you, we fixed it.
>
> *“immaterial --> do you mean not significant?”*: yes.
>
> *"In §2.2 insert a reference to the Appendix where $M$ is appropriately described"*: We added a reference, thank you.

---

> > ### Comment · Reviewer_J2FM · 2021-11-28
> > **Confirmed Score**
> >
> > Thank you to the authors for having addressed lot of my concerns but I decided to confirm my score.
> >
> > Even though the formalism has been changed and now the paper is more coherent and clearer, there isn't a statistically significant difference among tested approaches. Some interesting behavior emerges and for this I think that the paper is slightly above the acceptance threshold..
> > Moreover, the models related to the missing references are not tested. The authors decided to include the comparison to the camera ready version but I think it is too late.

---

### Official Review · Reviewer_jCBE · 2021-10-31

**Correctness:** 3
**Technical Novelty And Significance:** 3
**Empirical Novelty And Significance:** 2
**Recommendation:** 5
**Confidence:** 4

**Main Review:**

1. Experimental results
- Based on Table 1 and 8, the overall performances are very similar with SAINT.
- Definitely, the performances are not "statistically significant" better than SAINT.
- What are the main advantages of the proposed method in comparison to SAINT? It seems like in terms of the performances, the advantages seem very marginal and we cannot claim that the proposed method is better than SAINT.
- Also, based on Table 1, it seems like the differences from XGBoost and GBM are also marginal. In that case, are there any advantages to using deep learning models instead of those traditional models?

2. Standard deviations
- It is always good to show the standard deviations of the performances.
- We can see it in Table 1 (left) but not in Table 1 (right).
- Based on the marginal performance improvements, it would be more important to demonstrate this standard deviation.

3. Figures.
- It would be good if the authors provide a figure that overview the proposed networks.
- As the authors mentioned in the discussions, it would be more informative if the authors show some tSNE analyzes about the learned representations.

4. Intuitions
- The authors showed various results and ablation studies to discover the rules of thumbs for different cases.
- However, the intuitions for those conclusions are not enough. For instance, why is wide more effective than deep?

5. Extensions of the model
- It seems like the outputs do not need to be a categorical variable.
- In that point of view, can we extend this method to the regression problem?

**Summary Of The Paper:**

- The authors proposed a novel deep learning model for tabular data classification.
- The authors utilize the field-wise factorized networks to extract the meaningful information from both categorical and numerical features of the tabular data. Then, using those embeddings to solve the classification problem.
- The experimental results showed that the proposed method achieved comparable performances in comparison to SOTA deep learning models for tabular data (SAINT).

**Summary Of The Review:**

Strength:
- The authors proposed a novel and well-grounded deep learning model for tabular data classification problem.
- The authors provided extensive experimental results to understand the performances of the proposed method.

Weakness:
- The experimental results are not that promising. The performances are similar (not noticeably better) than SOTA. If the authors can claim the unique advantages of the proposed method in comparison to SOTA, I think I will increase my scores.
- The intuitions of the proposed method are limited.

-----------------------------------------

Thank you for the detailed answers from the authors.
I carefully read the authors' responses and other reviewers' reviews.
Overall, most reviewers are concerned on the weakness of the experimental results. This is also my main concern about this paper.
Although the authors claimed that the proposed method is "faster" than another baseline, it does not mean that this method is "faster" than other alternatives (we can easily find other traditional methods that are faster than the proposed method with similar performance).
For XGBoost and GBM, the authors said that those are not suitable for CTR prediction due to the high cardinality, However, the authors did not show how they are working in those kinds of datasets in the classification problem. If that claim is valid, it would be much better to show some datasets with that characteristic and show the failure of XGBoost and GBM.
Based on these, I am going to stay on my original score (5) because the performance concerns are not well resolved.

---

> ### Author Response · Authors · 2021-11-22
> **Response to Reviewer jCBE**
>
> We thank the reviewer for the dedicated review and for finding our method “novel and well-grounded”. The reviewer raised several valid points, which we believe can be addressed by the following clarifications.
>
> **Experimental results**
>
> ***“What are the main advantages of the proposed method in comparison to SAINT? It seems like in terms of the performances, the advantages seem very marginal and we cannot claim that the proposed method is better than SAINT.”***: SAINT suggests a very complex attention mechanism, as well as deep encoder layers (for the row model), while we show that theoretically grounded, simpler architectures can achieve similar results.  In addition, by comparing the runtimes between the models we can observe that SAINT inference times are much slower. Our theoretical analysis established an important  focus for tabular architectures -  wide fieldwise factorized nodes and a proper activation function. These fundamental although simple aspects were not considered in previous architectures, and can gain robust and fast results.
>
> *Runtime comparison between our method and SAINT on three tabular datasets.*
>
> |         | Income | Forest | Bank  |
> |---------|--------|--------|-------|
> | SAINT-s |  6.890 |  7.621 | 5.764 |
> | SAINT-i |  1.502 |  3.327 | 1.348 |
> | SAINT   |  2.449 |  4.480 | 2.107 |
> | F2NN    |  0.660 |  0.933 | 0.575 |
>
>
> ***“Also, based on Table 1, it seems like the differences from XGBoost and GBM are also marginal. In that case, are there any advantages to using deep learning models instead of those traditional models?”***: The advantages of deep learning models are by their ability to handle both high cardinality categorical fields, as well as numerical fields, without requiring further feature selection or processing mechanisms, which might cause loss of information. As an example, CTR prediction dataset, which contains high cardinality categorical fields, are not suitable for tree-based models. On the other hand, FM-based models require categorical fields, so numerical values are discretized. Our framework is very general and able to handle a wide range of tabular tasks and constraints. In addition, deep learning models allow representation learning, as well as an end-to-end learning with other modalities, which is a common use-case. Meaning, we achieve further gains by product of deep learning models.
>
>
> **Standard deviations**
>
> ***“It is always good to show the standard deviations of the performances. We can see it in Table 1 (left) but not in Table 1 (right). Based on the marginal performance improvements, it would be more important to demonstrate this standard deviation”***: The reported results are taken from one run, so there is no std in this case. However, the difference between different seeds is minor. As a demonstration, we ran in the last days our model and deepFmFM, the best baselines, on the Avazu dataset on a different seed (our reported experiments were on seed=42, now we changed to seed=43). We obtained $78.693$ for our model and $78.648$ for deepFmFM. We will run the entire comparison on both of the datasets and complete this requirement for the camera ready version.
>
>
> **Figures**
>
> ***“It would be good if the authors provide a figure that overviews the proposed networks.”***: We added an illustration to the appendix.
>
> ***“As the authors mentioned in the discussions, it would be more informative if the authors show some tSNE analyzes about the learned representations.”***: We think that the learned representations should be explored more carefully in order to get a better sense of their semantics and meanings. For this reason, we suggested it as future research.
>
>
> **Intuitions**
>
> ***“The authors showed various results and ablation studies to discover the rules of thumbs for different cases. However, the intuitions for those conclusions are not enough. For instance, why is wide more effective than deep?”***: The wide structure is derived by the theoretical analysis, such that every projection matrix of each field, $U_F$, projects a low-dimensional vector to a high-dimensional space. In our empirical experiments, we showed that given a wide layer, deepening the network with more layers of the same width does not improve the performance. Typically, deep neural networks are more narrow, but we need the width - both according to the theoretical background and empirically. However, as we mentioned in the practical consideration section, we do not claim that it is impossible to gain by using deeper networks, but that this is not trivial. Regarding the rule of thumb for the activation function - we presented a theoretical and empirical analysis, supporting GELU as an adaptive activation.

---

> > ### Comment · Reviewer_jCBE · 2021-11-29
> > **Thank you for the detailed answers from the authors.**
> >
> > I carefully read the authors' responses and other reviewers' reviews.
> >
> > Overall, most reviewers are concerned on the weakness of the experimental results. This is also my main concern about this paper. Although the authors claimed that the proposed method is "faster" than another baseline, it does not mean that this method is "faster" than other alternatives (we can easily find other traditional methods that are faster than the proposed method with similar performance). For XGBoost and GBM, the authors said that those are not suitable for CTR prediction due to the high cardinality, However, the authors did not show how they are working in those kinds of datasets in the classification problem. If that claim is valid, it would be much better to show some datasets with that characteristic and show the failure of XGBoost and GBM. Based on these, I am going to stay on my original score (5) because the performance concerns are not well resolved.

---

### Official Review · Reviewer_EGGP · 2021-11-02

**Correctness:** 2
**Technical Novelty And Significance:** 3
**Empirical Novelty And Significance:** 2
**Recommendation:** 3
**Confidence:** 3

**Main Review:**

The paper, in general, is well written, however the structure (in particular the RW and problem setup) can be improved.  The contributions, especially on studying and investigating the variants of F2NN (wrt architecture and activation functions) are interesting and seem solid.

strengths:
- an elegant method
- well documented ablation study
- well written paper

weaknesses:
- weak experimental setup
- weak (and potentially misleading) discussion of the findings

--
The major concerns I see, relate to the experimental setup wrt the comparisons to other methods and the reporting of the results. Namely, the authors report comparisons (on the classification problems) to several benchmark methods, typically applied to these tasks, but in reality they use results from a previous paper. It seems that none of the other methods have been tuned nor their hyper-parameters reported. Many of these benchmark methods are sensitive to their parameters, and their performance may improve significantly when properly used. On the other hand, F2NN, and one of the methods (SAINT) seem to be properly evaluated. This may lead to over-optimistic results, and misleading conclusions.

Another issue is how these results have been reported. The authors report mean performance, averaged over all 16 tasks (without reporting even std). This can also be misleading, since these are tasks with different properties, and the performance of different methods may substantially vary between them. IMO, analyzing and reporting them separately, not only will be more correct, but it also may provide other insights related to the performance of F2NN (which is given in Appendix C, but not discussed). Moreover, while I enjoyed the ablation study, the GELU adaptivity experiments seem a bit incomplete. The authors simulate the experimental setting reported by Rendle (2010), which is ok to begin with, but IMO should have investigated this further wrt other embedding-sizes and architecture designs (since one of the contributions is the GELU adaptivity).

There are several 'deep' FM/FFM approaches highlighted in the related work. On the other hand, the authors evaluate on some such methods (implemented within a public repo) but never discuss nor connect these two. Are some of the methods discussed (eg DeepFM) used in the experiments (eg. DeepFmFM)? If so, which ones and can you provide more details on them? Also for instance the one of Guo et al (2017) seems to be (partially) approximated by F2NN, but not really evaluated. How does F2NN compare to it?

The related work can also be improved in order to better serve the motivation for the work. Namely, personally, I don't see how the 'critiques of DL' paragraph fits, since this work doesn't seem to really support/nor dispute the stated claims. On a similar note, IMO the relation to Qi et al (2017) in the last paragraph, as it reads now - seems a bit of a stretch.

Considering, the 'devil is in the details' section, there is a statement regarding the benefits of the network depth for transformers. It is unclear, whether the referred experiments (that dispute this) concern F2NN or the transformer-based models, since it seems SAINT was tested with the same architecture, and TabTransformer not evaluated.

**Summary Of The Paper:**

The paper proposes a neural network architecture, inspired from factorization machines, for classification task from tabular data. In particular, the proposed factorized neural networks (F2NN) leverage the capabilities of shallow networks for learning factorized low-level representations that can approximate (the typically) high-dimensional representations of each field. This,in turn, together with different choices of activation factions, provides a generalization over different factorization approaches, and can lead to good (albeit not always better) performance on different classification tasks. Results from several (16) experiments on binary and multi-class classification tasks and two CTR tasks, comparing F2NN to different benchmark algorithms, show that it has comparable performance to some of standard (SOTA) methods.

**Summary Of The Review:**

The paper, in general, is well written, however the structure (in particular the RW and problem setup) can be improved.  The contributions, especially on studying and investigating the variants of F2NN (wrt architecture and activation functions) are interesting and seem solid. However, I have some concerns regarding the experimental set-up as well as the discussion of related work, which are reflected in my score.

---

> ### Author Response · Authors · 2021-11-22
> **Response to Reviewer EGGP**
>
> We thank the reviewer for the dedicated review and for recognizing our contributions as interesting and solid. The reviewer raised several valid points, which we believe can be addressed by the following clarifications.
>
> ***“It seems that none of the other methods have been tuned nor their hyper-parameters reported... On the other hand, F2NN, and one of the methods (SAINT) seem to be properly evaluated. This may lead to over-optimistic results, and misleading conclusions”***:
> * We tuned our two main and immediate competitors: the best attention-based models - SAINT variations, and the best FM-based models, resulting in improvements on their initial results. Our current results show better performance on the CTR prediction datasets, and competitive or slightly better results on general tabular datasets.
> * We agree that a careful tuning for tree-based models can be an important insight, we would be able to complete this extensive tuning for the camera ready version. However, please note that our current results show that our model is robust and competitive with tree-based models, while benefiting from being a neural model. Besides its ability to automatically handle use-cases that trees struggle with, such as high cardinality of categorical fields, having a robust neural alternative for tree-based models is much needed. As such, our model can be trained in an end-to-end manner with networks that operate on other modalities, as well as leveraging the byproduct learned representations for further applications.
>
> ***“The authors report mean performance, averaged over all 16 tasks (without reporting even std). This can also be misleading...”***: We do report the std of our runs, as well as of SAINT models. We also provided the results per dataset in the appendix. We do not have the std of the copied results, and therefore could not report them.
>
> ***“ IMO, analyzing and reporting them separately, not only will be more correct, but it also may provide other insights related to the performance of F2NN (which is given in Appendix C, but not discussed). “***:  Tabular data contain a variety of tasks, from a variety of domains and data types. The idea of evaluating on many dataset is to show robustness over this variety and to identify common trends. Discussing the results separately per dataset will not necessarily provide a wider insight, and therefore we did not analyze the results in this manner. What we stated in our analysis is the trend that can be observed also per dataset - “our performances are slightly better or comparable to SAINT variations, except from on multiclass tasks, where SAINT-s achieves poor performance.” The main insight is regarding the architecture - “While we use a set of linear projections that are only summed and passed through activation and classification layers, SAINT-s uses a massive six layer Transformer. This highlights that for robust and accurate neural models for tabular data, it does not appear that complex methods have an edge over well-tuned simple models such as ours.”  Besides the performance, we also conducted a further experiment, comparing the runtime between our method and SAINT. It can be seen that another benefit from our model is a faster evaluation time.
>
> *Runtime comparison between our method and SAINT on three tabular datasets.*
>
> |         | Income | Forest | Bank  |
> |---------|--------|--------|-------|
> | SAINT-s |  6.890 |  7.621 | 5.764 |
> | SAINT-i |  1.502 |  3.327 | 1.348 |
> | SAINT   |  2.449 |  4.480 | 2.107 |
> | F2NN    |  0.660 |  0.933 | 0.575 |
>
>
> ***“ the GELU adaptivity experiments seem a bit incomplete … IMO should have investigated this further wrt other embedding-sizes and architecture designs”***: The idea behind the GELU experiment was to demonstrate its effect by fixing an architecture and modifying only the activation function. In this manner, we are able to isolate and understand the effect of each activation function. Therefore we did not compare between different architectures.  However, we did explore it with respect to two different embedding sizes, the minimal (16) and the maximal (192) from Rendle (2010).
>
> ***“There are several 'deep' FM/FFM approaches highlighted in the related work...the authors evaluate on some such methods but never discuss nor connect these two. Are some of the methods discussed used in the experiments?”***: Among the FM-based models, FmFM is the leading shallow model, and DeepFwFM is the leading deep model, and is stronger than deepFM. Therefore we included these models in our comparison, and our model is indirectly better than the previous models such as DeepFM/FFM. We also experimented on a deep version of FmFM, demonstrating that our model is better even on the deep version of the best shallow model. Please note that there is no paper for DeepFmFM, this model was taken from the public code repo of FmFM.
>
> **Further responses can be found in the next comment**

---

> > ### Author Response · Authors · 2021-11-22
> > **Response to Reviewer EGGP - Cont.**
> >
> > ***“I don't see how the 'critiques of DL' paragraph fits, since this work doesn't seem to really support/nor dispute the stated claims”***: Our work supports the critiques on deep learning by showing that neither deeper transformers nor deepening our architecture improve the performance. In addition, we rely on Rendle (2010)’s insights on ReLU networks, and explore the effect and importance of different non-linearities.
> >
> > ***“ IMO the relation to Qi et al (2017)... seems a bit of a stretch.”***:  Qi et al (2017) is a well known work in the deep learning community. It was added to demonstrate how the theoretical analysis results in an architecture that shares ideas with a popular architecture, while making adjustment to the tabular context.
> >
> > ***“There is a statement regarding the benefits of the network depth for transformers. It is unclear, whether the referred experiments concern F2NN or the transformer-based models, since it seems SAINT was tested with the same architecture, and TabTransformer not evaluated.”***: The statement is related to the two mechanisms - (1) a network depth (2) an attention mechanism. Transformers usually implement these two mechanisms, but they were mentioned with respect to the attention mechanism. Either way, the experiments supports this statement in multiple ways:
> > * SAINT-i implements an attention mechanism on the dataset rows, SAINT-s on the dataset columns, and SAINT on both rows and columns. Our architecture obtains comparable or slightly better performance from all these variations, demonstrating that the attention mechanism is not beneficial in that case.
> > * SAINT-s is essentially a 6 layers transformer, demonstrating that both attention and depth do not improve the performance over our wide and simple architecture. Please note that SAINT-s is a better performing transformer than TabTransformer, therefore we did not conduct an experiment directly on TabTransformer.
> > * F2NN architecture does not benefit from adding layers of the same width, demonstrating that given a suitable width, there is no gain from deepening the network.

---

> > > ### Comment · Reviewer_EGGP · 2021-11-29
> > > **response**
> > >
> > > Many thanks to the authors for clarifying some of the issues I raised and modifying the manuscript accordingly. I also read the other reviews as well as the authors' comments to those reviews.
> > >
> > > However, as I stated in my original review, the comparison to SAINT (and the different variants) wasn't the issue, but rather the comparison to the other 9 benchmark methods, that were (and still are) reported incompletely. IMO these experiments could have been run (rather swiftly) in the rebuttal period, since most of these are standard, well-documented and available approaches (eg. the ensembles).
> > >
> > > re: the motivation of not analyzing/reporting the datasets separately: While I can see why the authors chose to report the average ROC-AUC (across datasets), I respectfully disagree that this, in this case, is better. Especially when these are problems with different sizes, properties and target different tasks --  with the evaluation mechanisms not reported (was it single train/val/test or repeated?) and not even the stds (across datasets) are reported. And please note here, I'm referring to the _other_ baselines reported, not SAINT. For instance, the authors state that there is difference between the performance of SAINT on binary vs. multi-class problems -- does this also holds for the other baselines (and to what extend)? With this in mind is hard to conclude something else that doesn't concern the comparison to SAINT.
> > >
> > > re: GELU ablation: As I stated in the main review, I understood and indeed found the GELU ablation important, but incomplete. I also understand why 16 and 192 were originally tested, but I find the authors' repose (to why not exploring other embedding sizes) rather unsatisfactory. For instance, the reasons for having K=192 and not larger since, there seems to be some improvement, are not really clear (other than being the one used in Rendle et al. (2020)*, which seem to also be chosen arbitrary to match and extend the experiments from NCF [1]).
> > >
> > > Having all these things in mind, I will keep my original score.
> > >
> > > --
> > >
> > > [1] He et al (2017) Neural Collaborative Filtering.
> > >
> > > *In the authors' response and the paper, the reference is Rendle(2010), but I think it is typo and it should be Rendle et al. (2020) "Neural Collaborative Filtering vs. Matrix Factorization Revisited" - since it is in this paper where these things are being evaluated.

---

### Official Review · Reviewer_c4rf · 2021-11-03

**Correctness:** 3
**Technical Novelty And Significance:** 4
**Empirical Novelty And Significance:** 3
**Recommendation:** 6
**Confidence:** 2

**Main Review:**

This theoretical connection between FM-based models and field wise neural networks is impressive. The generalization next to it is convincing and solid.

The line "M and is made symmetric" should be typo. The equivalence of upper triangular M and symmetric M should be stated explicitly.

The experiment shows quite marginal performance improvement. Do the authors have comments? This paper may pose new directions to improve current popular models given the theoretical connection. Any preliminary thoughts? the conclusion section doesn't include next steps. Increasing lowered rank in factorization can be explored.

In Appendix A, it said "FM, FwFM and FmFM, learn C vectors for each feature", should be "FM, FwFM and FmFM, learn C vectors, one for each feature"?

**Summary Of The Paper:**

This paper presents a general neural network framework for tabular data classification, that can recover a range of popular classification methods. The main components are per-field representation by factorized network and an activation function on aggregated representations of all fields. Experiments on general tabular classification and CTR prediction demonstrate its effectiveness, although performance gain is marginal.

**Summary Of The Review:**

The paper provides new insights on a unified neural network framework that can cover a range of popular tabular classification methods. The derivation is dense but solid. Although performance gain is marginal from experiments, the framework itself deserves more attention. It would be even better to pose new directions to improve models further.

---

> ### Author Response · Authors · 2021-11-22
> **Response to Reviewer c4rf**
>
> We thank the reviewer for the dedicated and positive review. We are glad the reviewer found our theoretical connection impressive and recognized that our framework deserves more attention. The reviewer raised several valid points, which we believe can be addressed by the following clarifications.
>
> ***"The equivalence of upper triangular M and symmetric M should be stated explicitly.״***: Thank you for pointing this out. The equivalence is in the quadratic form $x^TMx == x^T(\frac{1}{2}(M+M^T))x$. To improve clarity, we can change the definition so that M would be a symmetric matrix directly:  there is no order between the fields, for each pair only one matrix is learned, meaning $M_{FiFj} == M_{FjFi}$. Since it doubles the final result, we divide it by 2. Therefore the decomposed matrix can be written in the following form:
>
>
> \begin{equation}M= \frac{1}{2} \cdot
>   \begin{bmatrix}
>   0_K & M_{1,2} & M_{1,3} & ... & M_{1,J} \\\\
>   M_{1,2} & 0_K & M_{2,3} & ... & M_{2,J} \\\\
>   . & . & .& ... & . \\\\
>   . &.  & .& ... & .\\\\
>   . &. & .& ... & .\\\\
>   M_{1,J} & M_{2,J} &M_{3,J} & ... & 0_K
>   \end{bmatrix}
>   \end{equation}
>
> ***“The experiment shows quite marginal performance improvement. Do the authors have comments?”***:
> * **Accuracy**: In terms of neural architectures, our results show that none of the current approaches are able to gain superior results - neither multilayer perceptrons nor transformer encoders, as well as attention mechanisms. In addition, since we generalize FM-base models, which are the SOTA for CTR prediction and user-item recommendation, our framework should be the first consideration for these problems.
> * **Runtime**: We also examined the runtime between our model and SAINT. We can observe that using a transformer slows inference without an improvement to the performance.
> * **Cardinality and adaptivity**: Tree-based models might be robust models, though they cannot handle high cardinality datasets well, and also they cannot be trained together with other neural architectures (for example - train an image with its metadata). This is a very common use-case as neural architectures prevail in other modalities.
> * **Theory**: Our framework is theoretically grounded, and establishes the focus for tabular architectures - wide fieldwise factorized nodes and a proper activation function. These two fundamental yet simple aspects were not considered in previous architectures, and can gain robust and fast results.
>
> *Runtime comparison between our method and SAINT on three tabular datasets.*
>
> |         | Income | Forest | Bank  |
> |---------|--------|--------|-------|
> | SAINT-s |  6.890 |  7.621 | 5.764 |
> | SAINT-i |  1.502 |  3.327 | 1.348 |
> | SAINT   |  2.449 |  4.480 | 2.107 |
> | F2NN    |  0.660 |  0.933 | 0.575 |
>
> ***“This paper may pose new directions to improve current popular models given the theoretical connection. Any preliminary thoughts? the conclusion section doesn't include next steps.”***: We believe that ample evidence exists that improved regularization and hyper-parameter selection methods hold the key for increasing the performance.
>
> ***“In Appendix A, it said "FM, FwFM and FmFM, learn C vectors for each feature", should be "FM, FwFM and FmFM, learn C vectors, one for each feature"?”***:  Thank you for the correction, we fixed it.

---

> > ### Comment · Reviewer_c4rf · 2021-11-29
> > **Confirmed score**
> >
> > I have read author's rebuttal and other reviewer's responses. The main concern is still on its marginal performance improvement. The theoretical contribution is impressive to me, while I can also understand and accept if its board-line and got missed in the conference. Personally IMO, its slightly over acceptance threshold.

---

### Decision · Program_Chairs · 2022-01-20

**Decision:**

Reject

**Comment:**

The paper provides a deep learning technique aimed for tabular data via a unified view of factorization machines and other DNN approaches. The reviews are overall positive when discussing the provided technique, the motivation behind it and the writing. However, there are major concerns related to the experiments. The most dominant one is that of significance, meaning the advantage of the provided method when compared to existing literature. Other claims such as unclear details or different methods of reporting might be possible to resolve via minor edits, but this concern was not resolved in the rebuttal period. Before the paper can be published in a venue such as ICLR, it should provide a clearer comparison against previous works showing exactly where it improves upon them. At its current state, it doesn’t seem to be ready.